# Deciphering the Role and Signaling Pathways of PKCα in Luminal A Breast Cancer Cells

**DOI:** 10.3390/ijms232214023

**Published:** 2022-11-14

**Authors:** Emilio M. Serrano-López, Teresa Coronado-Parra, Consuelo Marín-Vicente, Zoltan Szallasi, Victoria Gómez-Abellán, María José López-Andreo, Marcos Gragera, Juan C. Gómez-Fernández, Rubén López-Nicolás, Senena Corbalán-García

**Affiliations:** 1Department of Biochemistry and Molecular Biology A, Veterinary School, Universidad de Murcia, CEIR Campus Mare Nostrum (CMN), 30100 Murcia, Spain; 2Instituto Murciano de Investigación Biosanitaria IMIB-Arrixaca, El Palmar, 30120 Murcia, Spain; 3Microscopy Core Unit, Área Científica y Técnica de Investigación, Universidad de Murcia, 30100 Murcia, Spain; 4Cardiovascular Proteomics and Developmental Biology Program, Centro Nacional de Investigaciones Cardiovasculares (CNIC), 28029 Madrid, Spain; 5Computational Health Informatics Program, Boston Children’s Hospital, Harvard Medical School, Boston, MA 02115, USA; 6Department of Bioinformatics, Semmelweis University, H-1092 Budapest, Hungary; 7Department of Cellular Biology and Histology, Biology School, Universidad de Murcia, CEIR Campus Mare Nostrum (CMN), 30100 Murcia, Spain; 8Molecular Biology Unit, Área Científica y Técnica de Investigación, Universidad de Murcia, 30100 Murcia, Spain; 9Centro Nacional Biotecnología, Consejo Superior de Investigaciones Científicas, 28006 Madrid, Spain; 10Department of Bromatology and Nutrition, Veterinary School, Universidad de Murcia, CEIR Campus Mare Nostrum (CMN), 30100 Murcia, Spain

**Keywords:** PKC, breast cancer, targeted therapy, kinases, signaling pathways

## Abstract

Protein kinase C (PKC) comprises a family of highly related serine/threonine protein kinases involved in multiple signaling pathways, which control cell proliferation, survival, and differentiation. The role of PKCα in cancer has been studied for many years. However, it has been impossible to establish whether PKCα acts as an oncogene or a tumor suppressor. Here, we analyzed the importance of PKCα in cellular processes such as proliferation, migration, or apoptosis by inhibiting its gene expression in a luminal A breast cancer cell line (MCF-7). Differential expression analysis and phospho-kinase arrays of PKCα-KD vs. PKCα-WT MCF-7 cells identified an essential set of proteins and oncogenic kinases of the JAK/STAT and PI3K/AKT pathways that were down-regulated, whereas IGF1R, ERK1/2, and p53 were up-regulated. In addition, unexpected genes related to the interferon pathway appeared down-regulated, while PLC, ERBB4, or PDGFA displayed up-regulated. The integration of this information clearly showed us the usefulness of inhibiting a multifunctional kinase-like PKCα in the first step to control the tumor phenotype. Then allowing us to design a possible selection of specific inhibitors for the unexpected up-regulated pathways to further provide a second step of treatment to inhibit the proliferation and migration of MCF-7 cells. The results of this study suggest that PKCα plays an oncogenic role in this type of breast cancer model. In addition, it reveals the signaling mode of PKCα at both gene expression and kinase activation. In this way, a wide range of proteins can implement a new strategy to fine-tune the control of crucial functions in these cells and pave the way for designing targeted cancer therapies.

## 1. Introduction

Breast cancer is the most commonly diagnosed cancer and the leading cause of cancer death among women worldwide [1]. Nowadays, breast cancer is a pathological disease that requires ongoing studies to obtain early prognostic and better treatments. Several signaling proteins are promising targets to confront breast cancer malignancy, and Protein Kinase C (PKC) family could be a potential candidate for designing new therapies [2].

Although PKC was identified 40 years ago by Nishizuka et al., like proteolysis-activated kinases [3], it was not until the discovery of PKC activation by tumor promotor phorbol ester that they turned into a new target in cancer studies [4]. This family has more than 60,000 citations in PubMed, and more than 10,000 citations are related to cancer. PKC family consists of at least ten members encoded by nine genes in mammals, which play essential roles in multiple signaling pathways such as cell proliferation, survival, and differentiation [5]. Notably, it has been impossible to establish whether PKCα isoform acts as an oncogene or tumor suppressor.

Mammary gland differentiation processes regulate the localization and expression of PKCs, and the overexpression of some PKCs in breast cancer has also been described [6]. Moreover, PKCs regulate several processes related to this disease, such as apoptotic and mitogenic signals [7]. Several reports indicate high PKC expression levels in this type of pathology, indicating the use of PKC as a therapeutic target, especially in the case of classical PKCα [8,9]. Approximately 8% of 1084 patients with invasive breast carcinoma showed alterations in PKCα, and 6% of them (67 cases) correspond to amplifications of this gene (cBioportal: https://www.cbioportal.org/) (accessed on 1 May 2022) [10,11].

The role of PKCα in tumor growth and its involvement in tumor progression is well known [12,13,14]. Particularly in breast cancer, the overexpression of PKCα confers a more aggressive phenotype. For example, cell lines that are Estrogen Receptor (ER)-positive become ER-negative in hormone-dependent breast cancer cells [15,16,17]. MDA-MB-231 cells, an example of triple-negative breast cancer cells, also exhibit high expression of PKCα, providing them with an enormous proliferative, migratory, and invasive capacity [18]. In addition, PKCα has also been found in the epithelial–mesenchymal transition process and confers high invasive and motility capacity to breast cancer cells [7,18,19].

To date, a wide variety of PKC inhibitors exist, and, despite numerous clinical trials in cancer, most of them have failed as therapeutic tools [20]. Since the scientific community understood how vital the phosphorylation events are in different signaling pathways, the activity to identify their dysregulation as hallmarks of cancer processes and the search for specific inhibitors has not ceased [21]. Imatinib is the first drug developed targeting a specific protein kinase (ABL tyrosine kinase). This kinase, expressed in nearly all cases of chronic myeloid leukemia (CML), paved the way for the design of new medicine based on the clinical development of kinase inhibitors [22]. Most kinase-targeted cancer therapies use multiple kinase inhibitors selected after screening and detecting critical markers in the patient’s tumor [23]. Many kinases have been considered potential targets for anticancer therapy [24]. Currently, some of the most important therapies inhibit the signaling of EGFR [25], PI3K/AKT/mTOR [26,27], CDK4/6 [28], or AURKA [29].

This strategy provides many examples of beneficial clinical applications but also faces many challenges and limitations due to the resistance that cancer cells develop to chemotherapy or the different response of patients to the same therapy. Therefore, the new therapies seek to combine the simultaneous inhibition of several kinases acting in the same pathway (vertical inhibition) or complementary pathways (horizontal inhibition) [30]. Recently, interest in developing so-called “personalized medicine” is leading to the creation of computational models combining multiple drugs to attack different targets and, at different times, improve patient treatment [31].

All this knowledge and experience led us to think about the possibility of systematically using a two-step strategy to inhibit PKCα expression. First, to slow down the tumor process and then to survey the signaling pathways affected. Second, to use the knowledge obtained to apply a set of specific inhibitors to modulate the activity of these cancer cells. Thus, we studied the effect of inhibiting PKCα expression using siRNA on cellular processes such as proliferation, migration, or apoptosis in a luminal A breast cancer cell line (MCF-7). Here, we explore the differential gene expression and kinase arrays to compare PKCα-Knock-Down (PKCα-KD) vs. PKCα-Wild Type (PKCα-WT) MCF-7 cells. Metascape functional enrichment analysis was used to ascertain the most critical pathways that were down- and up-regulated. Together with a phospho-kinase array, we identified a set of down-modulated signaling pathways that correlated well with controlling the growth and migration capacity observed in these cells. Analysis of the up-regulated pathways helped us select a set of targeted inhibitors to test whether their effect was synergistic with that of PKCα. We found that the inhibition of the PLC or tyrosine kinase pathways strongly affects proliferation and migration, further counteracting the tumorigenic phenotype.

## 2. Results

### 2.1. Knock-Down of PKCα in MCF-7 Cells Affects Their Tumorigenic Capacity

We wanted to elucidate whether the reduction in PKCα protein level influenced the tumor phenotype of this breast cancer cell line. Therefore, PKCα was knocked down in the MCF-7 breast cancer cell line using specific siRNA. We used a Western blot to measure the protein expression level with a specific anti-PKCα antibody (Figure 1A). The results show inhibition of PKCα expression of more than 55% from day 3 to day 8, reaching the maximum inhibition (70%) between days 4 and 6. During the same period (8 days), PKCα-KD cells showed a 48% reduction in proliferation on day 6 that persisted till day 8, indicating the role of PKCα in the proliferation process (Figure 1B).

As tumor expansion capacity, migration was also measured in PKCα-WT and compared to PKCα-KD MCF-7 cells by wound healing assay (Appendix A). A 61% reduction in cell migration was observed in PKCα-KD cells after 48 h, indicating that downstream phosphorylation by PKCα might be a critical step in reorganizing the cell cytoskeleton. PKCα-KD showed a slight increase in apoptosis (Appendix A). Overall, the results indicate that PKCα depletion reduces the tumor phenotype mainly by decreasing the proliferation and migration of this breast cancer model.

### 2.2. Transcriptomic Analysis of PKCα Knock-Down MCF-7 Cells Reveals Essential Alterations in Signaling Pathways

We interrogated the molecular pathways altered in MCF-7 breast cancer cells by decreasing PKCα protein expression. Thus, transcriptomic analysis was studied, and the mRNA expression transcripts of PKCα-WT and PKCα-KD MCF-7 cells were compared. A total of 13.461 unique mRNA transcripts were analyzed and the datasets were represented with the help of a volcano plot (Figure 2A). More than 5.000 probes were significantly different between the two cell types, displayed above the horizontal line (*p* < 0.05). When comparing PKCα-KD vs. PKCα-WT, 292 probes showed 1.5-fold up-regulation, whereas 248 probes showed 1.5-fold down-regulation (Figure 2A). Several genes distributed at different positions of the volcano plot were used for microarray validation by RT-qPCR (Figure 2A,B). The down-regulated *PRKCA*, *EGFR*, and *ITGβ6* genes, as well as up-regulated *PRKAR2B* and unaffected *SPAM1*, were validated (Figure 2B and Appendix A). Some genes (*ERBB4*, *PDGFA*, and *PLCB4*) showed low expression differences when validated by RT-qPCR, probably due to differences in cDNA synthesis and amplification methods compared to microarray.

We used Metascape (https://metascape.org/) accessed on 24 March 2022 [32] to perform the functional enrichment analysis of differentially expressed genes (DEGs) to better understand the functions and metabolic pathways after the knock-down of PKCα in MCF-7 cells. We studied up- and down-regulated DEGs separated into two sets for a more comprehensive analysis. As shown in Figure 3, the up-regulated genes group showed that the ten most overrepresented biological processes were: cellular response to hormone stimulus (GO:0032870), protein phosphorylation (GO:0006468), regulation of cellular response to stress (GO:0080135), intracellular protein transport (GO:0006886), regulation of cellular localization (GO:0060341), pathways in cancer (HSA05200), neuron projection development (GO:0031175), RHO GTPase cycle (R-HAS-9012999), regulation of growth (GO:0040008), and signal transduction by p53 class mediator (GO:0072331). The eleven most overrepresented biological processes in down-regulated genes were: Interferon Signaling (R-HSA-913531), regulation of response to biotic stimulus (GO:002831), antiviral mechanism by IFN-stimulated genes (R-HSA-1169410), negative regulation of cell population proliferation (GO:0008285), immune response to tuberculosis (WP4197), regulation of peptidase activity (GO:0052547), ER-phagosome pathway (R-HAS-1236974), response to wounding (GO:0009611), non-genomic actions of 1,25 dihydroxy vitamin D3 (WP4341), modulation by symbiont of entry into the host (GO:0052372), and signaling by receptor tyrosine kinases (R-HAS-9006934).

### 2.3. PKCα Controls the Phosphorylation of Essential Ser/Thr and Tyr Kinases and Transcription Factors

To obtain further insights into the kinases affected by the lack of PKCα, we performed an indicative assay using a human phospho-kinase antibody array. We compared PKCα-KD vs. PKCα-WT MCF-7 cell lysates (Appendix A). The array showed 14 proteins with lower phosphorylation levels in the PKCα-KD cells (Figure 4). These included PDGFr and AKT isoforms, a group of tyrosine kinases such as Lck, Fgr, or Hck, different isoforms of the STAT and β-catenin transcription factors, as well as DNA damage-related proteins Chk-2 and p27.

Although most proteins experienced decreased phosphorylation, surprisingly, we found that ERK1/2 and p53 (S392) were phosphorylated at a higher level when PKCα was down-regulated by siRNA treatment (Figure 4).

To obtain a complete overview of the complex cellular signaling depending on PKCα in MCF-7 cells, we analyzed the interactions and connections between the differentially expressed genes and the kinases affected in the absence of PKCα. We analyzed the protein–protein interaction (PPI) network using the STRING database [33] and Cytoscape software [34] (Figure 5). The PPI network consisted of 502 nodes and 1139 edges. The Cytoscape tool MCODE [35] identified the highly interconnected proteins. Two clusters appeared: Cluster 1 (22 nodes and 222 edges) and Cluster 2 (17 nodes and 51 edges) (Figure 5A). Cluster 1 is composed of proteins whose expression decreased after PKCα inhibition. The functional analysis of both clusters using the ClueGO tool [36] (Gene Ontology (Biological process) and KEGG database) revealed that cluster 1 is related to immune system regulation through interferon signaling and antigen processing and preparation via MHC I (Figure 5B). Cluster 2 showed a complex interaction between proteins and kinases. The main pathways were related to the epidermal growth factor signaling and the regulation of glucose or nucleocytoplasmic transport in the cell. KEGG functions such as proteoglycans in cancer or thyroid hormone signaling pathways (Figure 5C) also appeared.

### 2.4. Potential Targeted Therapy by Using Specific Inhibitors of Key Signaling Pathways

This work indicates that suppression of PKCα induces the down-regulation of critical oncogenic pathways. However, this study also shows unexpected up-regulation of crucial genes and kinases (e.g., *ERBB4*, *PLCβ4*, *PRKAR2B*, and *PDGFA*) whose roles in cancer development and progression are crucial (Figure 3). Therefore, we propose a dual strategy involving drugs that inhibit some of the up-regulated signaling pathways in PKCα-KD cells as a mode of targeted therapy to augment the inhibitory effect already exerted by PKCα suppression.

Thus, we used four different commercial inhibitors: U73122 (PLC and PLA_2_ families), KT5720 (PKA, catalytic domain), BMS 599626 (ErbB receptor family, catalytic domain), and Imatinib (ABL, KIT, and PDGF tyrosine kinases). We measured their effect on proliferation, migration, and apoptosis in PKCα-KD vs. PKCα-WT MCF-7 cells (Figure 6).

Inhibition of the PLC family by U73122 impairs the hydrolysis reaction of PIP_2_ into IP_3_ and DAG [37]. The effect of U73122 on PKCα-KD vs. PKCα-WT MCF-7 cells was measured using concentrations at 1 and 10 μM (Figure 6A). The results showed more significant inhibition of the proliferation rate in PKCα-KD cells (88%) compared to 49% in the same cells in the absence of the drug and 42% in the presence of PKCα and drug on day 8 (Figure 6A; Appendix A). U73122 treatment also produced 89% inhibition in the migration of PKCα-KD cells at 10 µM concentration compared to the 53% produced in the control cells at 72 h (Figure 6B; Appendix A). U73122 only promoted significant apoptosis at 20 µM in PKCα-KD cells (Appendix A).

Treatment with KT5720 at 0.1 and 1 μM, to inhibit the catalytic domain of PKA, produced a slight decrease in the proliferation of PKCα-KD in addition to the absence of PKCα (Figure 6C; Appendix A). The addition of 1 μM KT7520 reduced the ability of cells to migrate (89%) compared to 53% in the absence of the drug after 72 h (Figure 6D; Appendix A). However, the KT5720 treatment did not affect apoptosis (Appendix A).

BMS 599626 is a selective and efficacious inhibitor of the ErbB receptor family and is used at a wide range of concentrations (0.1, 1, 5, and 10 µM). The proliferative activity of PKCα-KD cells decreased to 92% compared to 45% in the absence of the drug. However, the drug also induced a robust inhibitory effect (82%) in PKCα-WT cells, which implies only an extra 10% inhibition by combining the two strategies (Figure 6E; Appendix A). Combining the inhibition of PKCα and BMS 599626, we obtained 80% inhibition compared to control cells treated with the same drug concentration (Figure 6F; Appendix A). Regarding apoptosis, combinatory treatment hardly improved the results of each treatment; although, a high concentration of 20μM showed a slight difference between PKCα-WT and PKCα-KD cells (Appendix A).

We used Imatinib at two different concentrations: 1 and 10 µM. We observed that imatinib (10 µM) decreased the proliferation rate of PKCα-WT MCF-7 cells above 47% (Figure 6G; Appendix A). When this treatment was applied to PKCα-KD cells, the proliferation inhibition was 85% compared to the control cells (Figure 6G; Appendix A). Moreover, the effect of Imatinib on the migration capacity hardly decreased in PKCα-WT cells. On the contrary, a substantial inhibitory migration effect of 68% appeared in PKCα-KD cells (Figure 6H; Appendix A). Imatinib did not exhibit a significant effect on apoptosis (Appendix A).

In conclusion, the results indicate that inhibitors of the PLC and tyrosine kinase pathways are more appropriate to control the proliferation of PKCα-KD MCF-7 cells and to control migration, inhibitors of the PLC, PKA, and EGFR pathways seem to be more appropriate. Other more potent drugs that modulate the PLC and EGFR pathways in that direction should be applied to induce apoptosis.

## 3. Discussion

PKCα and other isoforms are potential targets for cancer therapy, and it is challenging to determine the specific role of each isoform for different cancer types. PKCα is considered a tumor promoter, but in recent years, reports show that it can also function as a tumor suppressor, and its activity regulates cellular processes such as apoptosis, proliferation, or migration [14,38,39,40]. This duality in PKCα activity reveals the importance of studying each protein in its specific biological context to understand its role. Our study focused on characterizing the role of PKCα in a specific subtype of breast cancer (Luminal A breast cancer) using MCF-7 cells as a model. The suppression of its protein expression using siRNA has allowed us to identify a set of cellular processes that could be affected in its absence and whose knowledge could allow us to modulate the activity of this cell type [41,42,43]. In this work, we have obtained an overview of the signaling pathways affected in MCF-7 cells after PKCα inhibition by combining the results of DEGs with the activation status of kinases and the application of the MCODE tool (major gene clusters) on the PPI network.

Following PKCα inhibition, we found that interferon signaling was one of the clusters affected. MHC class I isoforms (*HLA-A*, *HLA-B*, *HLA-C*, and *HLA-F*) are involved in antigen presentation to cytotoxic CD8+ T cells and activate the adaptive immune system response [44,45]. Down-regulation of MHC class I has been described in many cancers as a strategy of tumor cells to avoid the immune system response [46]. Another group of down-regulated proteins participating in the interferon signaling pathway is the family of 2′-5′-oligoadenylate synthetases (OAS), consisting of *OAS1*, *OAS2*, *OAS3*, and *OAS-like* (*OASL*). High expression of these proteins correlates with better overall survival in basal-like breast cancer [47]. However, high expression of OAS1 and OAS3 was significantly associated with worse overall survival in Luminal A breast cancer [48]. Further studies will elucidate whether the low expression of these genes for a long time can cause resistance to therapies or evasion of the immune system. The results indicate that, in the short term, the tumor phenotype decreases in its ability to proliferate and migrate (Figure 1).

Proteins involved in wound response were down-regulated, and this would explain the low migration rate observed in wound healing assays of PKCα-KD MCF-7 cells. This group includes *ITGA2* and *ITGB4*, *KRT6A*, or *ELK3*, among others (Appendix A). The role of integrins in cell adhesion and their influence on cancer progression is well characterized [49]. Keratins are also diagnostic tumor markers; their low expression is a good indicator of the progression of the tumor [50]. At the same time, ELK3 expression correlates well with cell migration and invasion in the triple-negative breast cancer cell model (MDA-MB-231) [51], and the low expression of these proteins would favor the control of the cancer phenotype in MCF-7 cells.

One of the most common activated pathways in cancer processes is the phosphoinositide 3-kinase (PI3K)-AKT pathway, controlling cellular growth, survival, and key metabolic processes [52]. The PI3K-AKT pathway has been studied in breast cancer because it appears as one of the most affected signaling pathways by mutations, amplifications, and/or deregulation of proteins [53,54]. Our results showed that inhibition of PKCα affected PI3K-AKT signaling by controlling the expression of their regulatory subunits (*PI3KR1* and *PI3KR2*). These essential elements mediate the activity of the catalytic subunit by stabilizing it, inhibiting it, or allowing its interaction with other downstream elements [55,56]. The overexpression of PI3KR1 [57,58] and the inhibition of PI3KR2 [59] directly affect the catalytic capacity of PI3K, decreasing the PI3K-AKT signaling and tumor progression. The PI3K inhibition may result in the down-activation of AKT isoforms (AKT1 and AKT2), suggesting that PKCα positively regulates the PI3K-AKT signaling pathway.

Down-regulation of PKCα affected the expression of several proteins related to signaling by receptor tyrosine kinases (*ERLIN2*, *CDK2AP2*, and *TNS3*, among others). Most of them promoted breast cancer cell survival [60]; in addition, the kinase array also showed four tyrosine kinases with a lower level of phosphorylation, thus reinforcing the idea that PKCα also controls these critical pathways.

The STAT family is a significant element in this complex signaling [61,62]. Our data suggest that PKCα directly or indirectly controls the down-regulation of *STAT1* expression and phosphorylation in MCF-7 cells. Several reports showed that PKC isoforms activated these transcription factors in other cell lines [63,64]. The role of the STAT family in cancer is not straightforward, and depending on the cellular context, they function as oncogenes or tumor suppressors [65,66]. Elevated STAT1 levels are associated with therapeutic resistance in ER+ breast cancer cells [67]. Therefore, we could consider suppressing PKCα expression to a beneficial effect in this breast cancer cell line. The same situation occurs with other transcription factors such as *SOX2* and *SNAI2* that were also down-regulated. Both have been described as essential mediators for tumor progression and proliferation and provide endocrine resistance to conventional anticancer therapy [68,69].

Most of the signaling pathways inhibited due to PKCα down-regulation indicated that this kinase controls key signaling hubs in MCF-7 cells. Its inhibition correlates well with the results observed in proliferation and migration assays (Figure 1). The only pathways that alerted our attention were interferon-related since their downregulation correlates with disease progression and poor prognosis.

Analysis of the pathways regulated by PKCα inhibition also showed a high expression of the insulin-like growth factor type 1 receptor (*IGF1R*) that correlates with a favorable prognosis in hormone receptor-positive breast cancer [70].

*SMAD5* appeared up-regulated but not Aurora kinase A (AURKA), which cooperates to promote chemoresistance in breast cancer therapies [71]. In addition, we identified Aurora kinase binding protein (*AURKAIP1*) up-regulated. This protein is a direct activator of AURKA degradation, and thus, it is responsible for the pathway inactivation [29]. All these results correlate well with those obtained in our work regarding growth inhibition and cell migration.

ERK1 and ERK2 are master regulators of different pathways, such as cell proliferation, survival, growth, metabolism, migration, and differentiation [72]. Different stimuli trigger the ERK signaling through receptor tyrosine kinases (RTKs), G-protein-coupled receptors, or integrins, in addition to small GTPases Ras and Rap [73]. MAPK kinase signaling is a major culprit in breast cancer. This pathway is also responsible for tumorigenesis and cancer progression, and there are different drugs to inhibit their signaling [74]. However, many studies have shown good prognostic features associated with increased activation of MAPKs in ER+ breast cancers [75]. Numerous studies suggest that PKCα activates ERK signaling [76,77,78,79]. However, our data showed that ERK1/2 experienced increased activation after PKCα inactivation and correlated with decreased proliferation and migration capacity of MCF-7 cells. These results could support the idea that ERK activation mediated by PKCα inhibition has an antitumor effect in ER+ breast cancer.

One of the most important tumor suppressors in cancer research is p53 due to its role in inducing growth arrest or apoptosis after DNA damage or alteration in the cell cycle process, besides controlling other cellular mechanisms such as metabolism [80]. Many processes in the cellular context contribute to determining the decision between cellular survival or death, where p53 acts take the final jury’s verdict [81,82]. Our results showed an increase in p53 phosphorylation and up-regulation of several proteins related to this pathway after PKCα inhibition, supporting our previous results where PKCα-KD MCF-7 cells showed less capability to proliferate compared to PKCα-WT MCF-7 cells. p53 activation could be determined indirectly through AKT activity modulation [83] or direct activation mediated by a lack of PKCα [84,85].

A large variety of unexpected genes (*ESR1*, *ERBB4*, *PLCβ4*, *PDGFA*, *PRKAR2B*, *IGF1R*, *SMAD5*, or *MAPK3K2*, among many others, see Appendix A) appeared up-regulated. This fact is an essential finding after applying a PKCα inhibition strategy since, in the short term, the signaling balance indicated the inhibition of the progression toward the tumor phenotype.

The estrogen receptor ESR1 dimerizes and translocates to the nucleus upon ligand binding. It controls the transcription of many genes involved in cell cycle regulation, DNA replication, cell differentiation, or apoptosis. ESR1 also remains in the cytoplasm to interact with other proteins, such as receptors or kinases, participating in various signaling pathways [86]. Overexpression of the ESR1 gene potentially reduces the sensitivity of breast cancer cells to endocrine therapies, leading to disease progression and metastasis [87]. Studies showed that PI3K inhibition increases ER activity by regulating the histone methyltransferase KMT2D activity [88,89]. Likely, the observed inhibitory effect on the PI3K pathway due to the lack of PKCα promotes this unexpected compensatory effect.

Another up-regulated gene involved in protein phosphorylation and cancer pathways was *ERBB4*, while *EGFR* was down-regulated. EGFR is a significant target in cancer, as it is frequently mutated and/or overexpressed. Numerous drugs inhibit its activity [90]. Seven ligands connected to EGFR activators induce specific cellular responses and intracellular trafficking (Ras/MAPK pathway, the PI3K/AKT pathway, and the phospholipase C (PLC)/protein kinase C (PKC) signaling) [91]. In addition, other kinase-independent functions result from the heterodimerization of EGFR with other ERBB family members or interaction with kinases/phosphatases that trigger its signaling [92]. ERBB4 differs from the rest of the HER family members by undergoing alternative splicing and has four different isoforms depending on where the splicing site locates: the extracellular juxtamembrane region (JMa or JMb) or the cytosolic C-terminus, close to the tyrosine kinase domain (Cyt1 and Cyt2) [93]. Upon interaction with ligand, ERBB4 is activated and begins to phosphorylate its substrates, triggering different signaling pathways such as Ras-MEK-ERK or PI3K-AKT or promoting gene expression through its nuclear translocation [94]. Although ERBB4 plays an essential role in different types of cancer, its role as an oncogene or tumor suppressor depends on the cellular context, and it is still under study [95]. Also important is the endosomal recycling and degradation pathway that acts as the primary regulator of HER family expression and activation by controlling the availability of these receptors in the cell membrane and other cellular compartments [96]. The microarray results indicate decreased expression of the endosomal machinery. Numerous studies have described the role of PKC isoforms in regulating the HER family by controlling their travel from the membrane to endocytic vesicles. This traffic determines what vesicle proportion will be recycled and which one will degrade in the lysosome [97,98,99,100,101,102]. The results of proliferation and migration experiments indicate that the balance between EGFR and ERBB4 expression shifts towards inhibiting the tumor phenotype of MCF-7 cells.

The DEGs analysis also detected overexpression of the *PRKAR2B* gene encoding for the regulatory subunit of PKA. This result appears as a favorable prognostic marker in pancreatic and colorectal cancer and unfavorable in thyroid cancer. On the contrary, other work has shown that PRKAR2B depletion induces resistance to apoptosis, and in that sense, overexpression of the gene would be desirable [103,104].

Several genes, such as *PDGFA* or *PLCβ4* exhibited increased expression and are usually associated with poor prognosis in patients with solid tumors [105,106].

These latter results were significant in designing the rationale for selecting inhibitory drugs (Figure 6) to counteract the over-expression of these genes to push the balance toward a more inhibited tumor phenotype. This strategy demonstrated that inhibiting PKCα in the first step makes it possible to condition the MCF-7 cells into a low tumor profile due to the inhibition of critical cancer pathways (Figure 3 and Figure 4). In the second step, the specific strategy to inhibit unexpected up-regulated pathways allows us to further hinder this phenotype, with PLC and tyrosine kinases being the most important for proliferation and PLC, PKA, and EGFR being the most important for migration (Figure 6).

In conclusion, we have shed light on the role of PKCα in luminal A breast cancer cells. The integration of this information has allowed us to advance in two significant areas in cancer therapy. First, inhibiting a kinase that is not the leading cause of the tumorigenic effect is an appropriate strategy to curb the tumor phenotype. Second, to fully control the process, we can use differential expression and kinase activation to establish a personalized treatment based on the specific knowledge of the proteins affected. Implicitly, this research seeds the idea of a workflow for targeted therapy. Shortly, if perfected, it might be applied to a clinical performance protocol by testing in vitro this dual strategy on tumor cells extracted from the patient to precisely tailor the treatment.

## 4. Materials and Methods

Cell culture. MCF-7 breast cancer cells were purchased from ATCC and were cultured in Dulbecco’s modified Eagle’s medium (DMEM) supplemented with 10% FBS, 50 uds/mL penicillin, and streptomycin, 110 mg/mL pyruvate and 2 mM glutamine, in a 7.5% CO_2_ incubator at 37 °C.

siRNA transfection. PKCα expression was inhibited in MCF-7 cells using siRNA (Invitrogen, Waltham, MA, USA). In addition, a negative siRNA control was used in this experiment. The sequence used to inhibit PKCα expression was: 5′-UCCAAACGGGCUUUCAGAUCCUUAU -3′ (PKCα-KD). The negative control siRNA sequence was 5′-AGGUAGUGUAAUCGCCUUGUCGCCUUG-3′ (PKCα-WT).

MCF-7 cells were transfected with 1 µM siRNA (control non-target or inhibitory siRNA) using Mirus Ingenio^®^ Electroporation Solution (Mirus Bio LLC, Madison, WI, USA) by electroporation. Then, 3 × 10^6^ cells were resuspended in 300 μL of electroporation solution and added to 0.4 cm electroporation cuvettes (Bio-Rad, Hercules, CA, USA). Cells were electroporated in a GenePulser (Bio-Rad, Hercules, CA, USA) with two 900 V/0.4 ms square wave pulses and immediately placed on a fresh medium.

Western blot. Cells were lysed in lysis buffer (50 mM Tris HCl pH 7.5, 150 mM NaCl, 1% Nonidet P-40 (Roche Diagnostic, GmbH, Germany), 10% glycerol, complete commercial protease inhibitor, Mini, EDTA-free Protease Inhibitor Tablets and phosphatases inhibitor PhosSTOP (Sigma-Aldrich Chemistry, S.A., Madrid, Spain). After cellular lysis, cells were passed through a syringe (10×) and centrifugated at 14,000 rpm for 15 min at 4 °C. Electrophoresis loading buffer (200 mM Tris HCl pH 6.8, 250 mM DTT, 5% SDS, 37.5% glycerol, and 0.015% bromophenol blue) was added to supernatants and analyzed into polyacrylamide gel (SDS-PAGE). Proteins were separated and transferred to a nitrocellulose membrane (Hybond ECL (GE)). The membrane was blocked using blocking buffer (2% bovine serum albumin diluted in TBST (Tris-HCl 20 mM pH 7.5, NaCl 150 mM and Tween-20 0.1% (*v*/*v*)) for 2 h at room temperature. Then, primary antibodies were incubated O/N at 4 °C, and horse peroxidase-conjugated secondary antibodies were incubated for one hour at room temperature. WesternBright ECL-HRP substrate commercial kit (Advansta) was used to reveal the bands. The optical densities of Western blot bands analyzed with Fiji software [107] (https://imagej.net/software/fiji/downloads) calculated the percentage of PKCα inhibition and normalized by β-actin and using the level of each day PKCα expression as 100%. The bar plot shows the average percentage of inhibition per day ± standard deviation. Primary antibodies used were: anti-PKCα ab32376 (Abcam), anti-GAPDH ab9485 (Abcam and anti-β actin ab8227 (Abcam). Secondary antibodies used were: Goat F(ab’)2 Anti-Mouse IgG—F(ab’)2 (HRP), pre-adsorbed (ab5887) and Goat F(ab’)2 Anti-Rabbit IgG F(ab’)2 (HRP) preadsorbed (ab6112) (Abcam).

Compounds. U73122 and KT5720 were purchased from Sigma-Aldrich (St. Louis, MO, USA); BMS 599626 was purchased from Selleckchem (Houston, TX, USA), and Imatinib was purchased from Novartis (Basel, Switzerland). Compounds were dissolved in sterile DMSO, and they were used at the following concentrations: U73122 (1–10 µM), KT5720 (0.1–1 µM), BMS 599626 (0.1–10 µM), and Imatinib (1–10 µM).

Wound-healing assay. A wound-healing assay was used to evaluate the migration ability. Control and siRNA-transfected MCF-7 cells were seeded in 24-well plates to achieve a 90–95% confluence on the day of maximum inhibitory. The day before the assay, cell medium was replaced by low concentration FBS medium to synchronize the cellular cycle. The plates were washed with PBS after making a scratch in each well using a sterile pipette tip. The cultures were photographed until the wound was closed. In addition, different compound (previously described) was added to evaluate their effect on cell migration capability. Quantitative analysis of the wound area was carried out using Fiji [107] software from three independent wound-healing experiments.

Proliferation assay. Proliferation assay was assessed by fluorometric quantification of DNA using CyQUANT^®^ NF Cell Proliferation Assay kit (Invitrogen, Waltham, MA, USA), according to the manufacturer’s instructions. Briefly, control and siRNA transfected cells were seeded in 96-well plates (1 × 10^4^ cells) and allowed to attach for 24 h. Control and siRNA MCF-7 cell medium were replaced by medium containing drugs when appropriate. Then, the proliferation assay was measured for 8 days, where each condition was tested in three wells and three independent experiments at least.

Apoptosis assay. The percentage of apoptotic cells was determined by flow cytometry using the Vybrant apoptosis assay kit #4 (YO-PRO-1/propidium iodide; Molecular Probes/Invitrogen). Briefly, cells were washed twice with ice-cold PBS and resuspended in PBS containing YO-PRO-1/propidium iodide. Apoptotic cells were identified by flow cytometry after incubation for 20 min. Briefly, control and siRNA transfected cells were seeded in 96-well plates (1 × 10^4^ cells) and allowed to attach for 24 h. Control and siRNA MCF-7 cell medium were replaced by medium containing drugs where appropriate. Cells were treated until the day of maximum inhibitory. Then, apoptosis assay was measured, where each condition was tested in three wells and three independent experiments at least.

RNA extraction. RNA was extracted from cultured cells with the QIAGEN RNeasy pPlusMini Kit according to the manufacturer’s protocol. MCF-7 cells were transfected with control and inhibitory siRNA and RNA was extracted on the day of maximum PKC α expression inhibition. RNA concentration and purity were measured on a NanoDrop ND-1000 spectrophotometer (NanoDropTech, Wilmington, DE, USA).

Microarray. RNA isolated from PKCα-KD and PKCα-WT MCF-7 cells on the day of maximum inhibitory was analyzed using GeneChip^®^ Probe Array HG-U133 A2 (Affymetrix) according to the manufacturer’s instructions. Briefly, previously isolated RNA was amplificated and biotinylated using MessageAmp™ II-Biotin Enhanced Kit (ThermoFisher Scientific) according to the manufacturer’s protocol. Then, new biotinylated RNA was fragmented and hybridized with GeneChip^®^ Probe Array for 16 h at 45 °C. Chips were scanned using Affymetrix^®^ GeneChip^®^ Scanner 3000. Triplicates of each condition were performed. Microarray quality control was performed with PCA analysis (m) after data normalization with the RMA algorithm. Results were analyzed using Partek Genomic Suite (Partek Incorporated, St. Luis, MO, USA) software to determine genes with a significative different expression between both conditions (Appendix A).

Microarray data analysis. The transcripts selected using the above procedures were analyzed using Metascape web-based portal [32]. Metascape is a powerful web-based tool for gene annotation and gene list enrichment analysis that incorporates updated ontology databases such as KEGG Pathway, GO Biological Processes, Reactome Gene Sets, Canonical Pathways, and CORUM. To perform the analysis, all genes in the genome proceeded from the enrichment background. Terms with a *p*-value <0.01, a minimum count of 3, and an enrichment factor >1.5 are collected and grouped into clusters based on their membership similarities. In addition, the nodes that share the same cluster-ID are typically close to each other. The terminology used to name the transcripts is that approved by the HGNC.

RT-qPCR. RNA was harvested from transfected MCF-7 cells as described before and reverse transcribed using iScript Reverse Transcription Supermix kit (Bio-RadBio-RadLaboratories, Hercules, CA, USA) following the manufacturer’s instructions. The RT-qPCR was performed from cDNA templates using the SYBR Green PCR master mix and a 7500 fast real-time PCR system (Applied Biosystems). The transcript levels were determined after normalization against GAPDH and SPAM1, using the REST© 2022 software. All RT-qPCR experiments were repeated with at least three biologically independent replicates. Appendix A shows the list of the primers used in this study.

Protein–Protein Interaction (PPI) network analysis. In the present study, Protein–Protein Interaction (PPI) network of Differential Expression Genes (DEGs) from PKCα knock-down MCF-7 cells was constructed using the online database Search Tool for the Retrieval of Interacting Genes (STRING; http://string-db.org) (version 11.5) [33]. An interaction with a combined score >0.7 was considered statistically significant. The thicknesses of those edges were associated with the combined score. The STRING interaction network was imported into Cytoscape software (version 3.9.1) [34]. Cytoscape is an open-source bioinformatics software platform for visualizing molecular interaction networks and further analysis. Cytoscape app MCODE [35] was used to visualize the significant nodes and partition the network into different modules with degree cut-off = 2, cluster finding = haircut, node score cut-off = 0.2, K-score = 2, and maximum depth = 100, respectively. Functional enrichment analysis of identified clusters was performed by using ClueGo (version 2.5.8) [36] plugin in Cytoscape environment. The parameters used for ClueGo analysis were a two-sided (Enrichment/Depletion) tests based on the hypergeometric distribution for enrichment analysis. The *p* value  <  0.05 was corrected by Bonferroni step-down correction method. Only Gene Ontology (Biological Processes) and KEGG databases were used in the analysis. Terms that passed the *p* value threshold (*p*  <  0.05) were considered significantly enriched.

Proteome profiler human phospho-kinase array. The relative levels of protein phosphorylation were determined by proteome profiler human phospho-kinase array kit (Cat#: ARY003B, R&D Systems, Minneapolis, MN, USA). Protein extraction was performed as mentioned in Western blot assay. Briefly, 400 μg of each MCF-7 cell lysate (PKCα-KD and PKCα-WT) were incubated with the microarrays pre-coated with antibodies against 43 kinase phosphorylation sites at 4 °C overnight. After washing, the microarrays were incubated with biotin-labeled antibodies at RT for 2 h. Then microarrays were incubated with HRP-conjugated streptavidin at RT for 30 min on a rocking platform. Chemiluminescence and Density of spot pixels were analyzed using ImageJ software. Each membrane has its controls described in Appendix A, and the intensities are normalized with respect to the 100% signal (reference spot). Background signal from each spot was subtracted using the average signal from the negative control spots. Note that each kinase is tested in two independent spots, and several kinases of the same family are represented. Only one biological sample was used in this assay.

Statistical analysis. Statistical data analysis and visualization were performed with Estimation Statistics database (www.estimationstats.com) [108]. All data points were presented in a Cumming estimation plot to display the underlying distribution. The raw data are plotted on the upper axis. On the lower axis, mean differences are plotted as bootstrap sampling distributions (5000). Each mean difference is depicted as a dot. Each 95% confidence interval is indicated by the ends of the vertical error bars. An additional permutation test was also applied to confirm these results. The *p* values to accept/reject the null hypothesis of no differences between PKCα-WT and PKCα-KD cells are indicated below each comparison (CI 95%).

## Figures and Tables

**Figure 1 ijms-23-14023-f001:**
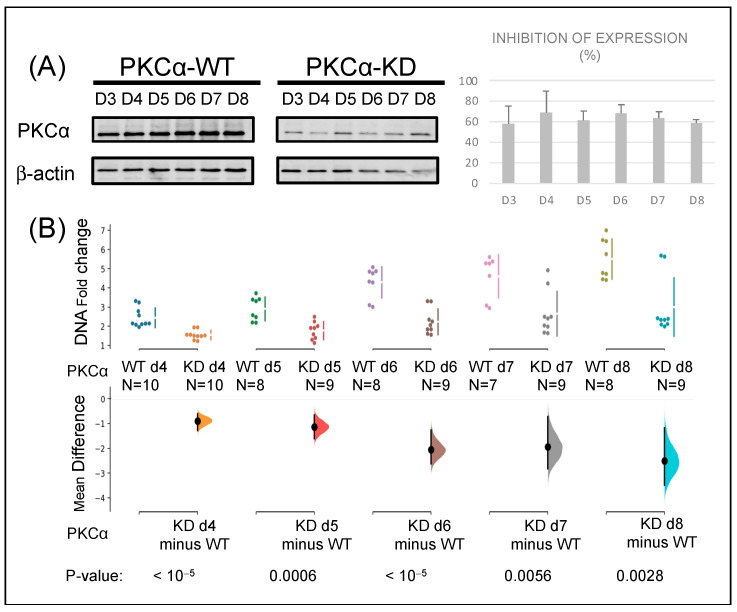
Knock-down of PKCα in MCF-7 cells affects their proliferation capacity: (**A**) MCF-7 cells were transfected with siRNA specific for PKCα. Protein expression was assessed by WB relative to β-actin expression at the days indicated. (**B**) Proliferation rates of PKCα-WT and PKCα-KD MCF-7 cells were measured by fluorometric DNA quantitation. The mean difference for two comparisons against the PKCα-WT and PKCα-KD cells are shown in the Cumming estimation plots (see Materials and Methods for a more extensive explanation).

**Figure 2 ijms-23-14023-f002:**
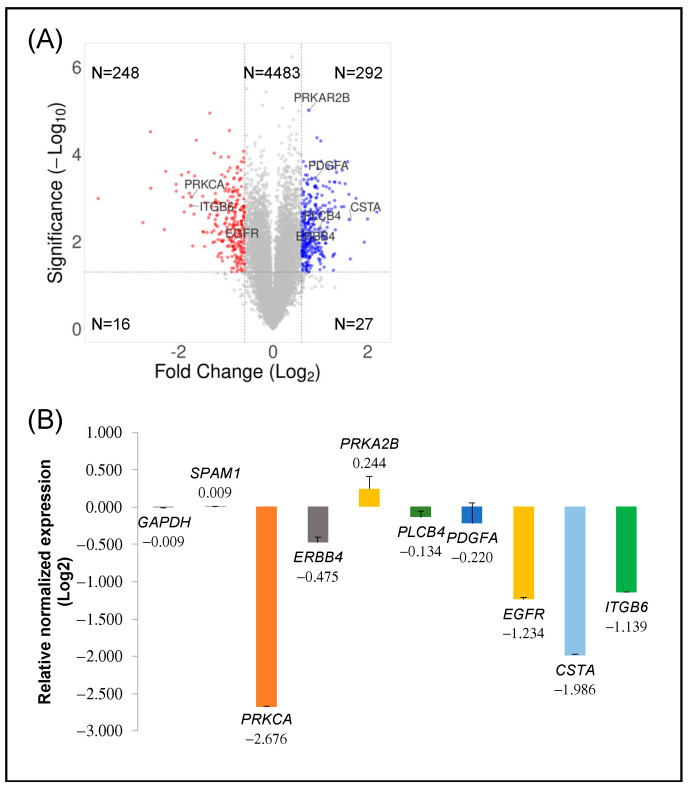
Transcriptomic analysis: (**A**) Volcano plot displaying the expression of probe sets in PKCα-KD vs. PKCα-WT MCF-7 cells. Log_2_ fold changes and their corresponding *p*-values of all genes in the microarray were taken to construct the volcano plot. Genes up-regulated with more than 1.5-fold change with a *p*-value <0.05 are depicted in blue dots, and those down-regulated with identical fold change and *p*-value are in red dots. All other genes in the array are not significantly altered and are represented in grey dots. (**B**) Validation of significant up-and down-regulated genes by qPCR.

**Figure 3 ijms-23-14023-f003:**
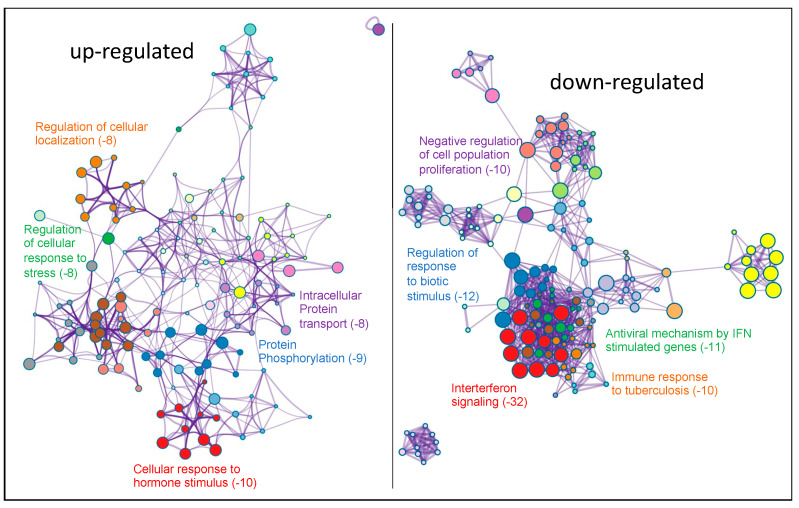
Functional enrichment analysis with Metascape of DEGs obtained after comparing PKCα-KD vs. PKCα-WT MCF-7 cells. Network of enriched terms colored by cluster-ID. DEGs on the left represent the up-regulated genes and on the right represent the down-regulated genes. Here, we show the five top clusters with enriched representative terms, and the numbers in brackets indicate the (Log_10_ (*p*-value)). Appendix A show the genes included in each cluster.

**Figure 4 ijms-23-14023-f004:**
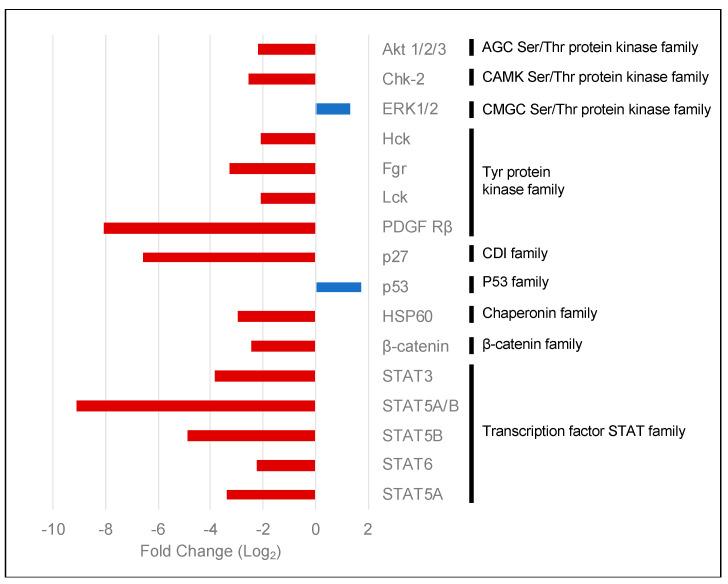
Human phospho-kinase array comparing PKCα-KD vs. PKCα-WT MCF-7 cells. The histogram shows the fold changes of the kinases affected by the down-regulation of the expression of PKCα in MCF-7 cells. Red bars represent phosphorylated proteins showing a decrease in phosphorylation levels, whereas blue bars show an increase.

**Figure 5 ijms-23-14023-f005:**
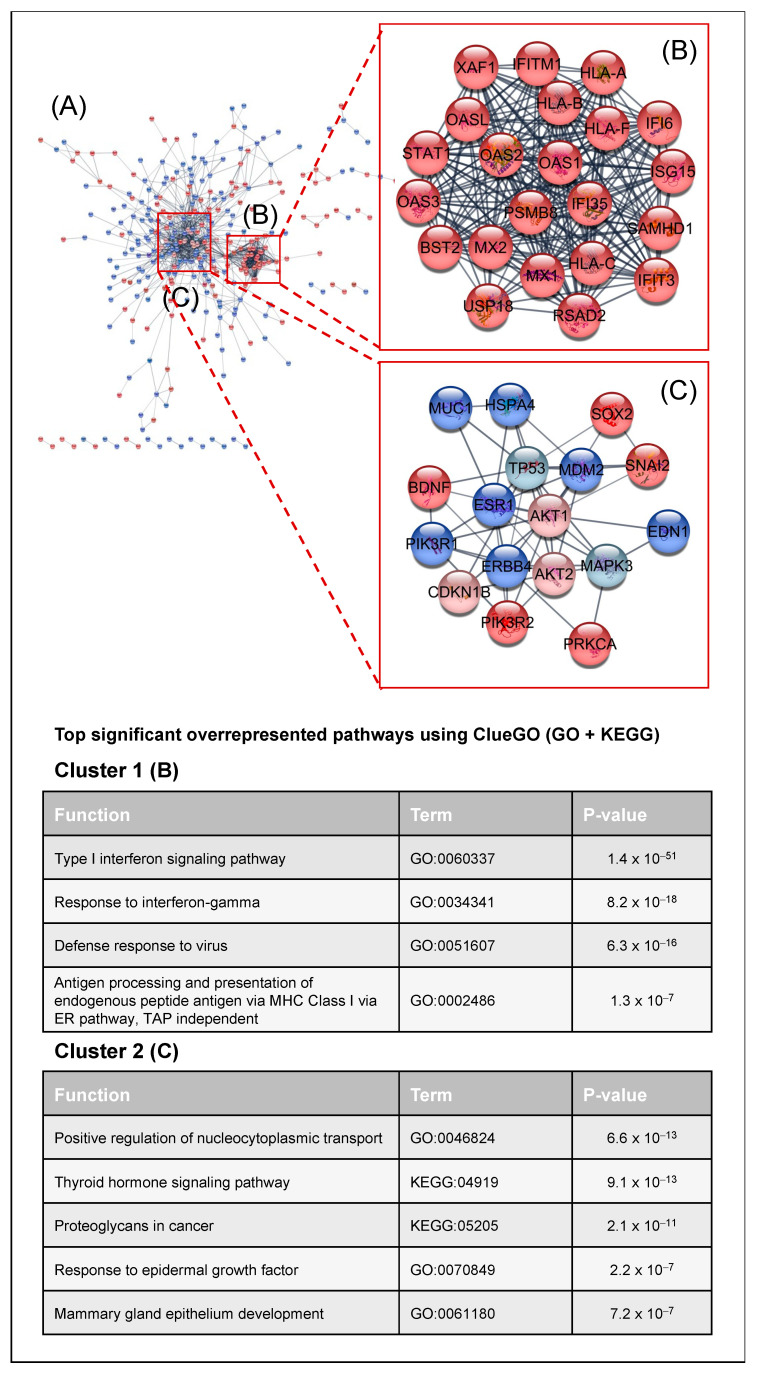
Critical signaling pathway analysis: (**A**) Protein–protein interaction network of DEGs using the STRING database [33]. DEGs obtained from the microarray analysis were colored in dark red when down-expressed, whereas up-expressed DEGs were colored in dark blue. Down- and up-activated kinases obtained from the phospho-kinase array were depicted in light red and light blue, respectively. The MCODE tool was used to identify the two main protein clusters present in the protein-protein interaction network: Cluster 1 (22 nodes and 222 edges) (**B**) and Cluster 2 (17 nodes and 51 edges) (**C**).

**Figure 6 ijms-23-14023-f006:**
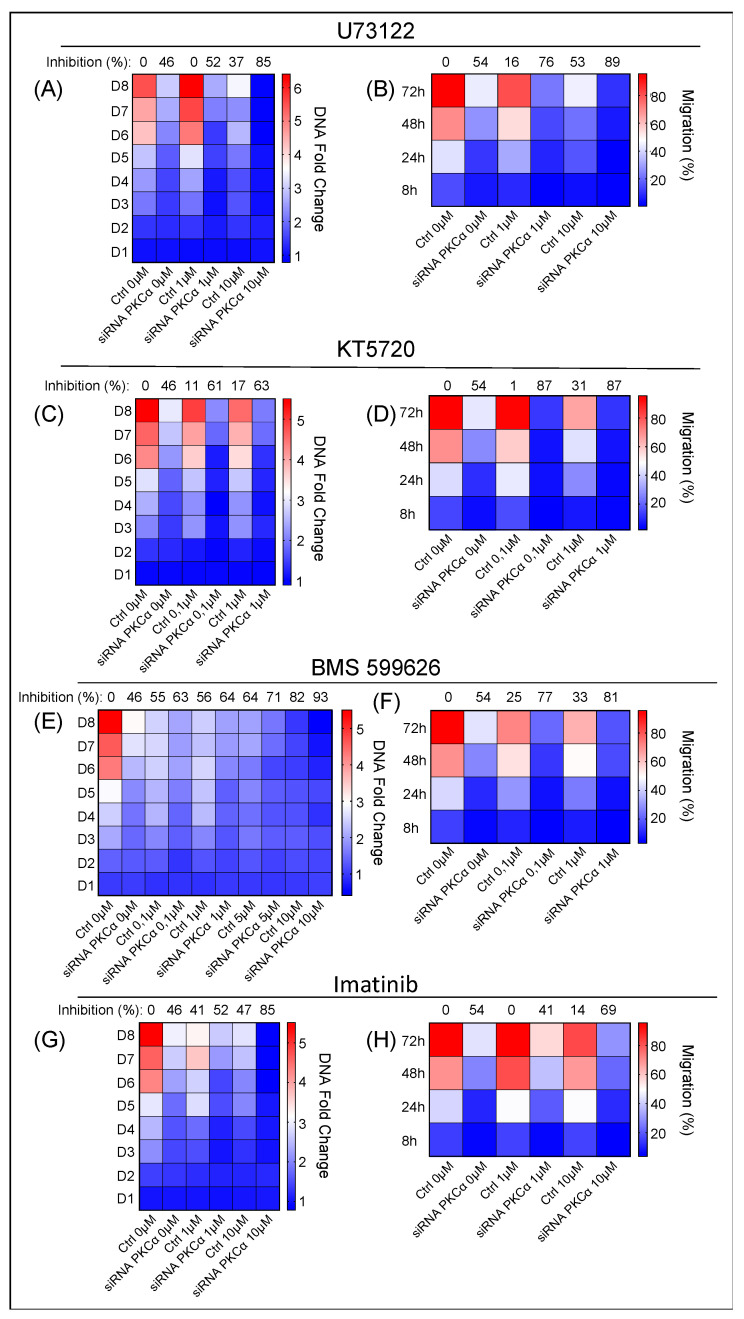
Effect of additional inhibition of the key signaling pathways identified in the arrays. The proliferation and migration capabilities of PKCα-WT and PKCα-KD MCF-7 cells were measured under different treatments: (**A**,**B**), U73122; (**C**,**D**), KT5720; (**E**,**F**), BMS 599626; (**G**,**H**), Imatinib on PKCα-WT and PKCα-KD MCF-7 cells. A heat map-type graph was used with the dark blue color corresponding to the lowest value for the DNA fold change or migration % and the dark red color to the highest value. The effect of each treatment on PKCα-KD vs. PKCα-WT MCF-7 cells at day 8 (proliferation assay) or 72 h (migration assay) is shown at the top of the heat map as the percentage of inhibition on the ability to proliferate or migrate.

## Data Availability

Microarray data are available in the ArrayExpress database (http://www.ebi.ac.uk/arrayexpress) under accession number E-MTAB-12374.

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
