# Peer review of "Deciphering the Role and Signaling Pathways of PKCα in Luminal A Breast Cancer Cells"

_ijms, 2022, doi:10.3390/ijms232214023_

Round 1

Reviewer 1 Report

Breast cancer is the most common cancer in women worldwide and the second most common cancer in the U.S. The expression of the estrogen receptor α (ER), progesterone receptor (PR) and HER2, as assessed by immunohistochemical method, are the basis on which the optimal treatments for patients were elaborated.

Precision medicine for cancer approaches diagnosis, treatment and prevention which consider the patient’s own genes and the genes or other markers carried by the cancer cells.

This manuscript studies the possibility of developing a therapy for breast cancer that takes into account the genes hyper-expressed or suppressed by the silencing of PKC-alpha. The main strategies were analysis in  Microarray of transcripts changes and PPI interaction network of Differential Expression Genes from PKC-apha knock down cell line and Proteome profiler human phosphokinase array .

This manuscript is potentially interesting, although the study is exclusively performed using an in vitro cell line, MCF7, and does not make use of patient’s  biopsies to validate expression level of PKC alpha and genes or signalling target of PKC-alpha that can be involved in proliferation invasiveness and tumour progression .

As a matter of fact, tumour tissue and blood are usually collected for analysis, often genetic. Then authors had the opportunity to take advantage of biopsies to compare their data about transcripts alterations in hyperexpressing PKC-alpha MCF7  with that of patients.

Since I made this review without any Supplementary data, entirely absent in the site of manuscript, I could not process a proper revision of the manuscript and I merely make a few observations about the data that did not strictly require Supplementary material.

Fig1  It needs a restructuring between A and B for better understanding?

Fig2B Table S1 missing

Fig3 Table S2 missing

FigS2 missing, this figure, showing  the phospho-kinase array of PKC-alpha KO vs PKC-alpha WT MCF7, should have been important for the evaluation of signalling alterations due to PKC alpha KO. The mere description is better when the experimental result  is also shown. The exposure of membrane array is very important and different exposure might result in different quantitative evaluatin. For that reasons more than one array must have  been  done for each sample .Authors did not mention how many times repeated the array/sample.

Table S3, S4 and S5 are missing .

Authors are invited to compare the main results from MCF7  with biopsies from patients.

Authors are also invited to submit all the material necessary for a correct revision by reviewers

Reviewer 2 Report

1.       Most kinase targeted cancer therapies are based on the use of multiple kinase inhibitors that are usually selected after screening and detection of key markers in the patient’s tumor. This should be elaborated.

2.       In present times, studies targeting kinase inhibitors is a on a high due to the fact that kinases govern the crucial cell cycle steps. A little more explanation regarding kinases, kinase inhibitors and a growing research of kinases inhibitors in anticancer therapeutics shall be included in the introduction to make it more solid.

https://doi.org/10.3390/biomedicines8050119

https://doi.org/10.3390/ijms222010986

3.       The results are presented nicely with novel findings.

4.       Figures are nicely depicted; the legends can be made crisper that will make it look more attractive.

5.       Focus on consistent usage of abbreviations throughout the manuscript.

6.       Avoid usage of some long sentences in the manuscript, using GRAMMARLY advanced version shall aid in the overall improvement of the English language of the manuscript. For instance

b) to continue with a second action, is necessary to characterize this effect at the level of differential expression and activated kinases to establish a specific personalized treatment based on the  specific knowledge of the proteins affected. This needs to be restructured for better understanding.
